# Syphilis as Re-Emerging Disease, Antibiotic Resistance, and Vulnerable Population: Global Systematic Review and Meta-Analysis

**DOI:** 10.3390/pathogens11121546

**Published:** 2022-12-15

**Authors:** Yaneth Citlalli Orbe-Orihuela, Miguel Ángel Sánchez-Alemán, Adriana Hernández-Pliego, Claudia Victoria Medina-García, Dayana Nicté Vergara-Ortega

**Affiliations:** Centro de Investigación Sobre Enfermedades Infecciosas, Instituto Nacional de Salud Pública, Cuernavaca 62100, Mexico

**Keywords:** re-emergence, syphilis, antibiotic resistance, CORE group, A2058G

## Abstract

Syphilis is a sexually transmitted disease that has become a public health problem, especially in vulnerable populations. A systematic review and time-free meta-analysis of the publications on the molecular detection of syphilis and mutations associated with antibiotic resistance, CORE group, and syphilis genotypes in PubMed databases, Scielo, and Cochrane was performed, and the last search was conducted in June 2022. Proportions were calculated, and standard errors and confidence intervals were reported for all results of interest. We included 41 articles for quantitative extraction and data synthesis. An increase was observed in the proportion of subjects diagnosed with syphilis and the presence of the A2058G mutation during the 2018–2021 period compared to 2006 (70% _95%_CI 50–87 vs. 58% _95%_CI 12–78), and we observed that the greater the proportion of the population participating in men who have sex with men (MSM) (<50% and >50%) syphilis increased (78% _95%_CI 65–90 vs. 33% _95%_CI 19–49). In conclusion, we suggest that there are a set of characteristics that are contributing to the resurgence of syphilis and the selective pressure of bacteria. The MSM population could be a vulnerable factor for this scenario and the global presence of A2058G and A2059G mutations that confer resistance to macrolides.

## 1. Introduction

Syphilis is a sexually transmitted disease (STD) caused by *Treponema pallidum* (*T. pallidum*). In the late 1990s, people thought that syphilis had disappeared. Years later, this stealth pathogen re-emerged as a public health problem, especially in vulnerable populations such as men who have sex with men (MSM), people living with HIV (PLWH), female sex workers (FSWs), male sex workers (MSWs), people deprived of liberty, and pregnant women [1,2,3,4].

The Centers for Disease and Control (CDC) in 2017 reported that 15–20% of people in the U.S.A. had a previous syphilis infection, and between 2013 and 2017, there was an increase in cases of up to 76%. According to the Health Data Organization, the estimated global prevalence is 49.7% (_95%_CI 38.3–66) per million cases of syphilis. For women, the estimated prevalence was 18.7% (_95%_CI 14.9–24.1) and 31% (_95%_CI 23.1–41.8) per million cases for men. It seems that the global burden of the disease relapses in the male population [5]. The World Health Organization (WHO) estimated 7 million new syphilis cases in 2020 [6].

The risk factors associated with acquired syphilis are sexual behavior, serosorting among PLWH, multiple sexual partners, the use of preexposure prophylaxis (PrEP) to compensate for HIV risk behavior, and social networking sites or mobile device apps to find sex partners [7]. In 2018, at least 33,927 cases of syphilis were reported in 29 European Union/European Economic Area member states by the European Surveillance System (TESSY), with a raw rate of seven cases per 100,000 population. The reported syphilis rates were nine times higher in men than in women, 29 cases per 100,000 population, and showed a peak age of onset of 25 to 45-years, in men [8].

Although syphilis is a mandatory reporting disease worldwide, the problem increases when it presents asymptomatically. It is believed that nearly 43% of people with acquired syphilis do not present with any clinical symptoms [9]. Due to these factors, the disease may be misdiagnosed. Many countries, including Mexico, have scarce information and reports. There is an urgent need to evidence the increasing number of cases and pay special attention to this old enemy, even more so in vulnerable populations [1].

The diagnosis of syphilis is based on clinical symptoms and confirmatory laboratory tests. There are treponemal tests (i.e., ImmunoSorbent-ELISA Linked Enzyme Assay) and nontreponemal tests (i.e., Venereal Disease Research Laboratory test-VDRL and Rapid Plasma Reagin-RPR). The nontreponemal tests (NT) have been routinely used because they are simple to operate, inexpensive, and rapid, although the sensibility is relatively low in the diagnosis of syphilis and is dependent on the stages (62–78% in primary syphilis, 97–100% for secondary syphilis, and 82–100% for early latent syphilis). In case of treponemal tests, the sensibility and specificity are better than non-treponemal tests (100% in secondary syphilis, 95.2–100% sensitive in early latent syphilis, and 86.8–98.5% sensitive in late latent syphilis). However, detection of antibodies by treponemal tests in the evaluation of treatment efficacy are difficult, and itis not possible to distinguish between active stage and previously treated infection [10]. Recently, the CDC approved the use of polymerase chain reaction (PCR) for testing of *T. pallidum* on suspected primary lesions, as one of the recommended tests. PCR specificity and sensibility have been high (96.6% and 78.4%, respectively) in cases of primary syphilis, with respect to clinical diagnosis, and better than serology, as it does not require the presence of antibodies or large numbers of bacteria for microscopy, and has been suggested to be a complementary technique in the detection of syphilis [11,12,13].

*T. pallidum* has been rising in recent years [14]. According to the WHO, use of macrolides and tetracycline as an alternative treatment for syphilis has increased [14]. This is especially true in people with allergies or in countries where penicillin is not easily accessible (i.e., Australia, Cambodia, New Zealand, and the Philippines) [15]. Along with increasing cases of syphilis, antimicrobial resistance (AMR) is emerging. Macrolide and tetracycline resistance has been the main focus of epidemiological surveillance. Punctual mutations such as A965T and G1058C, in 16S rRNA, and A2058G and A2059G, in 23S rRNA, have conferred *T. pallidum* resistance, even when treated with tetracyclines and macrolides. These last mutations are the most frequent and well-studied.

For these reasons, the aim was to systematically review and integrate epidemiological evidence, with respect to syphilis reemergence and the increase in AMR from A2058G and A2059G point mutations worldwide, as well as to quantitatively summarize evidence by conducting a meta-analysis.

## 2. Materials and Methods

We performed a systematic review and meta-analysis according to the PRISMA Declaration (Preferred Reporting Items for Systematic Reviews and Meta-Analysis) [16], that has the PROSPERO (International Prospective Register of Systematic Reviews) ID: CRD42022369676.

We searched publications regarding the molecular detection of syphilis, mutations associated with antibiotic resistance, core group, and syphilis genotypes in the databases PubMed (Public Medline): https://pubmed.ncbi.nlm.nih.gov/ (accessed on 25 June 2022), Scielo (Scientific Electronic Library Online) https://www.scielo.org/ (accessed on 25 June 2022), and Cochrane: https://www.cochranelibrary.com/cdsr/reviews (accessed on 25 June 2022).

Preliminary studies were selected using the following search algorithm: (“syphilis”, OR “*T. pallidum*” OR “*Treponema pallidum*” OR “prevalence of syphilis” AND “MSM” OR “Men sex men” AND “heterosexual” AND “pregnant” AND “PLWH” OR “people living with HIV” AND “penicillin” OR “penicillin g benzathine” AND “doxycycline” AND “azithromycin” AND “ampicillin” AND “macrolides” AND “tetracycline” AND “genotyping” AND molecular characterization” AND “mutations of resistance to antibiotics” AND “A2058G” AND “A2059G”. There was no time period restriction, and the last search was performed in June 2022.

Three authors screened (YCOO, AHP, and CVMG) evaluated independently (triple-blind) the titles, abstracts, and finally, the full texts of relevant studies according to eligibility criteria; duplicate reports were excluded. Discrepancies were resolved by the author DNVO.

Our search was based on the following research questions: (1) Is there an increase in syphilis worldwide? (2) Are MSM the core population group in syphilis transmission? (3) Is there published evidence about point mutations that confer antibiotic resistance in *T. pallidum* as a causal agent of syphilis? and (4) Is there an association between syphilis genotypes and antibiotic resistance? Taking into account the components of the PECOS declaration (population, exposure, comparators, results, and study design) [17], we applied the following eligibility criteria: (1) literature published in English and Spanish that was studied only in humans and on any date, (2) original observational studies (prospective and retrospective), (3) molecular detection of *T. pallidum* DNA and antibiotic resistance confirmed with PCR assay in the same article, (4) excluded gray literature, and 5) excluded abstract and full text not available.

### 2.1. Data Extraction

From eligible studies, information was extracted independently by the authors AHP, CVMG, and two other authors (YCOO and MASA) conducted the peer review, with discrepancies between information discussed by the four researchers. The following data were collected in a standard form for each article: first author and publication year; continent; study design; recruitment year; sample size; sex; age range or median/mean in years; a sample size of MSM population; sample size of PLWH; the proportion of molecular detection of syphilis; proportion of A2058G mutations; proportion of A2059G mutations; previous treatment; strain of *T. pallidum*; place where the participants were recruited; and the economic income of the country. In cases where data was missing in the studies, we communicated directly with the authors to request this information.

The quality assessment of each study was carried out independently from the Newcastle–Ottawa Scale (NOS) [18], modified to evaluate cross-sectional studies (Appendix A). Discrepancies in article quality scores were resolved by consensus, and a final grade was assigned.

The last author (DNVO) served as arbiter and made a final determination regarding the discrepancies in the data extraction and quality assessment.

### 2.2. Statistical Analysis

Crude proportion estimates were calculated, and standard errors (SEs) were reported for all outcomes of interest: the molecular detection of syphilis, the proportion of A2058G mutations, and A2059G of 23S rRNA *T. pallidum*. We use meta-analyses of the random effects model to pool estimates (with 95% confidence intervals [_95%_CIs], using the command “metaprop” in Stata, which can incorporate prevalence close to zero or one, using Freeman-Tukey double arcsine transformation [19]. The presence of heterogeneity between the selected studies was evaluated using the Cochran Q test and quantified with the I^2^ test (*p* value of <0.05 was defined as indicative of a statistically significant difference in results) [20].

Subgroup analysis was performed to identify sources of heterogeneity. The proportion of molecular detection of syphilis and A2058/A2059G were independently evaluated by reporting year, continent, the economic income of the country, and population type.

The proportion estimates of syphilis and A2058G/A2599G mutations were stratified over periods of years (2006–2012, 2013–2017, and 2018–2021) by continent (Europe, America, Oceania, Asia, and Africa) and were classified by the low, low-middle, upper-middle, and high economic income levels of the country, according to the World Bank [21], and the upper middle-income economic stratum was subdivided in studies that were conducted in China vs. studies that were not conducted in this country. In addition, they were classified by the characteristics of the population (MSM, FSWs, and PLWH). Since the majority of people were recruited in clinics for Sexually Transmitted Infections (STI), the remaining studies worked as unreported data. The MSM group was categorized as more than 50% (>50%) and less than 50% (<50%) of the total population that participated in each eligible study. Furthermore, to evaluate the proportion of A2058G and A2059G mutations, we stratified with the data of previous antibiotic treatment.

We assessed the potential presence of publication bias in the meta-analysis through visual inspection of funnel plots in search of asymmetric patterns and by Egger testing (*p* value < 0.10) [22].

All statistical analyses were performed with the Stata statistical package (version 14, release 2021; StataCorp, College Station, TX, USA).

## 3. Results

### 3.1. Characteristics of Study Eligible

Initially, 1449 articles were identified in the PubMed (*n* = 1412), Scielo (*n* = 2), and Cochrane (*n* = 35) databases, of which 490 duplicate articles were manually excluded. After reviewing titles and abstracts, 913 articles were excluded as they were not considered relevant to the objectives of this review. The full text of 46 articles was reviewed, excluding 5 articles that did not meet the *T. pallidum* determination by PCR. Finally, 41 articles were retained for quantitative extraction and synthesis of data (Figure 1).

The study characteristics are presented in Table 1. All eligible studies (*n* = 41) were cross-sectional in design; three studies were performed in Africa, thirteen in America, twelve studies were performed in Asia and Europe, and only one study was performed in Oceania. The sample sizes ranged from 26 to 3037 participants. In most studies, people of both sexes were included, but there was greater participation of male subjects compared to the female population (mean of 154 vs. 75, respectively). The age range was reported to be between 0.1 and 97.3 years. According to the characteristics of the population type, we observed that Coelho EC et al. exclusively included the FSW population [23]. Similarly, the study conducted by Mikalová L et al. exclusively included people living with HIV (PLWH) [24], and Liu D et al. indicated that the population included in their study was the general population [25]. Almost 61% (25/41) of the studies included a proportion of MSM in their total population. The population was recruited mostly in clinics specializing in sexually transmitted infections (STIs) (32/41, 78%); four studies reported that the population came from general care clinics, and two studies were performed in HIV care clinics (Table 1).

Fifteen articles reported data on previous antibiotic treatment (6 to 12 months before recruitment in the study). Two more studies reported reported that they excluded from their study the participants who taken antibiotics 12 months before, were excluded. In the remaining 24 studies, there was no report of these data. Approximately 25/41 (61%) of the articles reported *T. pallidum* genotyping, of which 19 reported a higher frequency of strain 14d. More than half of the studies (22/41, 53.7%) were classified within the high economic income level, 16 studies in the upper middle economic income, and 3 in the low economic income (Table 1).

All studies reported the proportion and/or the number of samples/patients with positive PCR of *T. pallidum*. The A2058G mutation of 23S rRNA *T. pallidum* was studied in 39/41 (95%) articles, of which only 36 identified this mutation. Only 25/41 studies analyzed the *T. pallidum* A2059G mutation, and 11 of them were able to amplify it (Table 1).

Most of the studies (37/41, 90%) used lesion samples for the molecular detection of T. pallidum and point mutations, and all studies reported patients with primary syphilis (Appendix A).

### 3.2. Bias Risk Assessment

According to the criteria of the NOS scale adapted for cross-sectional designs, 24/41 studies were of high quality, and 44% were of moderate quality (Table 2). There was no evidence of publication bias when using the Egger test (data not shown).

### 3.3. Meta-Analysis of Proportion of Syphilis and Antibiotic Resistance

We evaluated the proportion of molecular detection of syphilis from eligible studies (*n* = 41) and observed an overall proportion of molecular detection of syphilis of 66% (_95%_CI 54–74), with a heterogeneity of 99.25% (*p* < 0.001) (Appendix A). Therefore, the periods of the year (2006–2012; 2013–2017; and 2018–2021) were analyzed as possible variables involved in heterogeneity. In this meta-analysis, we observed an increase in the proportion of subjects diagnosed with syphilis over the year (58% _95%_CI 38–78; 66 _95%_CI 50–80; and 70 _95%_CI 50–87, respectively); the heterogeneity value for all groups (intra-groups) was I^2^ > 99%, *p* < 0.001 (Figure 2). Figure 2 shows the proportion of syphilis diseases stratified by year period (2006–2012; 2013–2017; and 2018–2021). The proportional overall was 0.65 (_95%_ CI 0.54–0.74) Estimates of the proportion (%) of the molecular detection of syphilis by period year. ES, Error Standard, CI, and confidence interval.

Subsequently, we stratified by type of population, and we observed that the higher the percentage of the MSM population, according to the total of participated in the study (<50% and >50%), the higher the rate at which syphilis increased (33% _95%_CI 19–49%; I^2^ = 99.03%, *p* < 0.001) vs. 78% (_95%_CI 65–90; I^2^ = 98.31, *p* < 0.001). Studies that did not report the type of population resulted in 72% (_95%_CI 54–88, I^2^ = 99.33%, *p* < 0.001) (Figure 3). Figure 3 shows the proportion of syphilis diseases stratified by type of population (percentage of MSM population participating in each study; they do not report (not reported) any characteristics of the population; General population; HIV population and FSW population). Estimates of the proportion (%) of the molecular detection of syphilis by type of population. ES, Error Standard, CI, and confidence interval.

We assessed the meta-analysis of resistance to specific groups of antibiotics (macrolides and tetracyclines), and we observed that there was only one report that identified resistance to tetracycline (data not included). However, there were no changes in the estimators when only A2058G mutations were included, so only analyses with macrolide resistance mutations were addressed (A2058G/A2059G).

The detected overall proportion of the A2058G mutation of 23S rRNA *T. pallidum* was 58% (_95%_CI 42–73, I^2^ 98.97%, *p* < 0.001) (Appendix A). When evaluating the presence of macrolide resistance mutations by periods of years, we realized that the increase in the A2058G mutation of 23S rRNA *T. pallidum* was proportional to the period (2006–2012: 39% _95%_CI 15–65; 2013–2017: 55% _95%_CI 27–81; 2018–2021: 76% _95%_CI 54–90), and intragroup heterogeneity was >97%, *p* < 0.001). (Figure 4). Figure 4 shows the estimates of the proportion (%) of the mutation A2058G by period year (2006-2012, 2013–2017, 2018–2021). The proportion overall was 0.58 (95% CI 0.42–0.73) ES, Error standard; CI, confidence interval.

When stratified by continent, Asia presented a greater proportion of A2058G mutations (76%; _95%_CI 41–98), followed by studies in Europe (61%; _95%_CI 38–81). The Americas showed a proportion of 52% _95%_CI 30–73) and the group of countries in the African continent presented a proportion of 4% (_95%_CI 0–23). We did not have intra-group proportions for Oceania since only 1 article was included. On the other hand, although few studies identified the A2059G mutation, we noticed proportions of this mutation in the countries belonging to America of 4% (_95%_CI 1–8) and Europe of 2% (_95%_CI 0–5) (Figure 5).

We analyzed the A2058G mutation according to economic income level and found a proportion of 70% (_95%_CI 58–81) in studies conducted in countries with a high-income level. In countries with a low-income level, the proportion was 4% (_95%_CI 0–23). Interestingly, we observed an increase in the A2058G mutation in studies conducted in China, compared with studies not performed in this country (73% _95%_CI 31–99) vs. 24 (_95%_CI 0–63). (Figure 6).

## 4. Discussion

In this work, we quantified the overall molecular detection of syphilis disease and the proportion of A2058/A2059 mutations that confer macrolide resistance from studies that met the eligibility criteria. The hypothesis of our work was based on the increase in the prevalence of syphilis and mutations conferring resistance to macrolides.

Although the characteristics of the studies and the complexity of the identification of antibiotic resistance mutations did not allow us to find the source of heterogeneity, we were able to recognize potential sources such as the participating MSM population, the yearly period of the report, continents, and/or income level of the country where the studies were done.

Several sources have reported a significant increase in cases of *T. pallidum* infection, which is consistent with our results, given that we identified an increase in the combined proportion of molecular detection syphilis in the period 2018–2021. According to the WHO in 2019, there were 6.3 million men and women aged 15 to 49 with infections caused by *T. pallidum* [61]; however, the CDC stated in the STI surveillance report in 2020 that syphilis cases increased by 7% compared to the previous year [62]. We observed an increase in the number of cases of syphilis, and therefore suggest that this disease could become a re-emerging disease.

Although the search for the studies did not focus on MSM, most included a significant percentage of that population. We noticed a higher combined proportion of syphilis in studies reporting a percentage of MSM participants greater than 50%, which suggests that this population could be the new core group of the disease. Chew et al. identified that the MSM population infected with *T. pallidum* increased significantly between 1999 and 2008; these authors indicate that this increase is due to high-risk sexual practices [63].

When analyzing the mutations of resistance, we observed a 50% increase in the combined proportion of the A2058G mutation in 2018–2021, compared to 2006–2012. A study conducted by Grimes et al. identified an increase in the A2058G mutation in samples from patients in San Francisco and Dublin over a period of 5 years [60], which is consistent with our results.

Similarly, the combined proportion was higher in studies conducted in Asia than in those conducted in other continents. This could be due to most of the studies having been conducted in China, and this approach was best observed when stratified by economic income, where the proportion of the A2058G mutation was higher in this country. China has a serious problem of antibiotic resistance; some studies estimate that more than 80% of antibiotics are prescribed in the diagnosis of influenza, in addition to being used inordinately in the food industry [59,64]. In contrast, a lower combined proportion was obtained in African countries; a study by Müller and collaborators suggests that the low proportion of the A2058G mutation in these countries is due to the low availability of macrolides [39]. Studies that identified A2059G had a very low frequency, which could be because this mutation was recently identified [65]. However, we can observe that in the opposite case, regarding the mutation A2058G, the countries with the highest combined proportion were those that belonged to the American continent, which suggests further screening of this mutation and further identification of characteristics of the population that could be associated with the point mutation A2059G of 23S rRNA *T. pallidum*.

In this work, it is important to note that this public health problem is not only the increasing rate of *T. pallidum* infection, but also identifying the central group that maintains transmission and could therefore be the cause of the spread of this disease. The second scenario focuses on the global epidemic of antibiotic resistance, in which syphilis disease has not been free of this scenario. Therefore, it is essential to have efficient tools for the identification, genotyping, and diagnosis of antibiotic-resistant *T. pallidum* and therefore, provide effective treatment against the disease and prevent the spread of strains resistant to this antibiotic.

According to the CDC’s STI treatment guidelines [15] alternative antibiotics to macrolides (including erythromycin and azithromycin) are recommended in 16 countries; the duration of treatment with this antibiotic is approximately two weeks for early syphilis and four week for late syphilis, but the evidence is limited for treatment efficacy of alternative treatment [15]. The latter are allowed only when penicillin is not available or when the patient is allergic to penicillin, or when trained personnel and injection equipment are scarce [15]. Currently, macrolides are highly recommended for the treatment of syphilis, because its activity in vitro against *T. pallidum*, reaches high levels in tissues and it has a long half-life [66], however, there is insufficient evidence of treatment failures and a high rate of *T. pallidum* chromosomal mutations associated with azithromycin resistance and other macrolides. So, this use is inconsistent with current CDC guidelines, which occurs more frequently outside the United States, which is a consequence of ease of administration (i.e., oral and single dose) or by alternatives for populations with specific characteristic (i.e., allergies, etc.) [12,13,63]. 

However, there is a variety of evidence showing that treatment with macrolides has more disadvantages than advantages. According to the CDC, Erythromycin and azithromycin should not be used in pregnant women as there is insufficient evidence of successful syphilis treatment [12]. In addition, there are genetic characteristics of the bacteria, such as the change from adenine (A) to guanine (G), at positions A2058 and A2059, in the 23S rRNA gene of *T. pallidum* [67,68] addressed in this study.

The limitation of this study is not being able to analyze whether there is a higher proportion of the A2058G/A2059G mutations associated with previous antibiotic treatment since most of the eligible studies did not contain these data (information was requested from the authors, but they did not have this); however, the meta-analysis was carried out and no difference was found between those who received versus those who did not receive previous treatment (data not shown). There are some other concerns about the development of antibiotic resistance, the use of Postexposure Prophylaxis (PEP), and Pre-Exposure Prophylaxis for STI Prevention (PREP), which have been reported [14]. Some studies have reported that administration of azithromycin and doxycycline as PEP or PREP demonstrated reduced gonorrhea, chlamydia, and syphilis [69,70,71]. However, further studies are needed to determine the effectiveness of the use of antimicrobials for PrEP or PEP STIs, and the potential impact on the development of antibiotic resistance [14].

In addition, the lack of information on *T. pallidum* genotyping (especially complete genotyping or use of 2–3 genes; probably due to the nature of the bacteria and the difficulty of its culture) did not allow us to address whether the strains are related to the presence of macrolide mutations. According to Marra et al., the increase in A2058G mutations could be due to the excessive use of macrolides in infections unrelated to syphilis and is significantly associated with a higher prevalence of resistant *T. pallidum* strains [72]. The strength of our work was the inclusion of eligible studies that performed only the direct detection of *T. pallidum* and antibiotic resistance mutations by molecular methods. Shukalek CB, et al. indicated that PCR had a specificity of 99.9% and sensitivity of 78.4% for the detection of *T. pallidum* [13], which allows an objective evaluation, since most studies confirmed the diagnosis of syphilis with molecular tests.

## 5. Conclusions

Syphilis can be diagnosed in a timely manner and is a curable disease; however, syphilis cases are increasing, leading to re-emergence. We note that most of the studies included a percentage of MSM in such a way that this population appears to be the new core group, and could be related to the spread of this public health problem. The global presence of mutations A2058G and A2059G in the 23S rRNA gene of *T. pallidum* (which confers macrolide resistance) could be the consequence of selection pressure in recent years, due to the excessive use of diverse antibiotics (not the first line of treatment for syphilis).

There are a set of characteristics that could be causing the increase of cases of this disease and bacterial evolution. In recent years, it has been shown that the trend in the use of indistinct antibiotics to penicillin contributes significantly to antimicrobial resistance. Although the evolution of *T. pallidum* is slow, we do not rule out its addition to the Global Antimicrobial Resistance and Use Surveillance Systems (GLASS) list by the WHO. Therefore, it is important to establish strategies that help to efficiently diagnose syphilis and help to have stricter surveillance of treatment, and prevent increased antibiotic resistance.

Although this work does not focus on any characteristic on the place of recruitment of the population, most of the eligible studies were conducted in STI clinics, of more information is needed on the dynamics of this disease in the general population.

## Figures and Tables

**Figure 1 pathogens-11-01546-f001:**
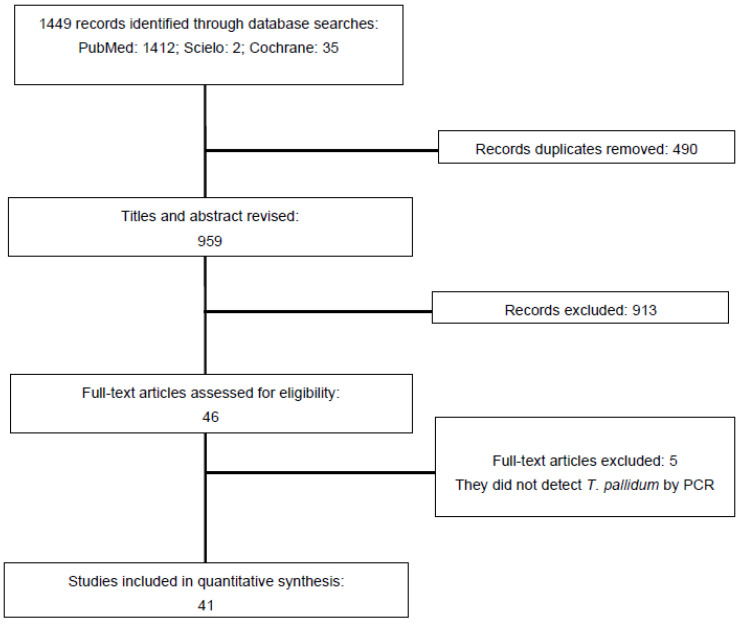
The PRISMA diagram statement for reporting systematic reviews and meta-analyses. Flow diagram of study selection of the 41 eligible studies according to the PRISMA statement reporting systematic and meta-analysis.

**Figure 2 pathogens-11-01546-f002:**
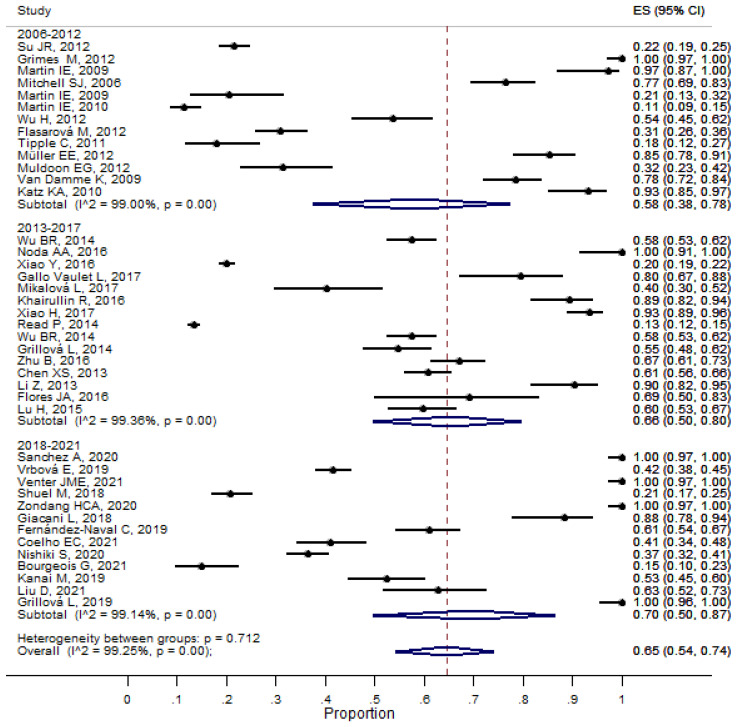
Forest plot of meta-analysis of the molecular detection of syphilis by year period.

**Figure 3 pathogens-11-01546-f003:**
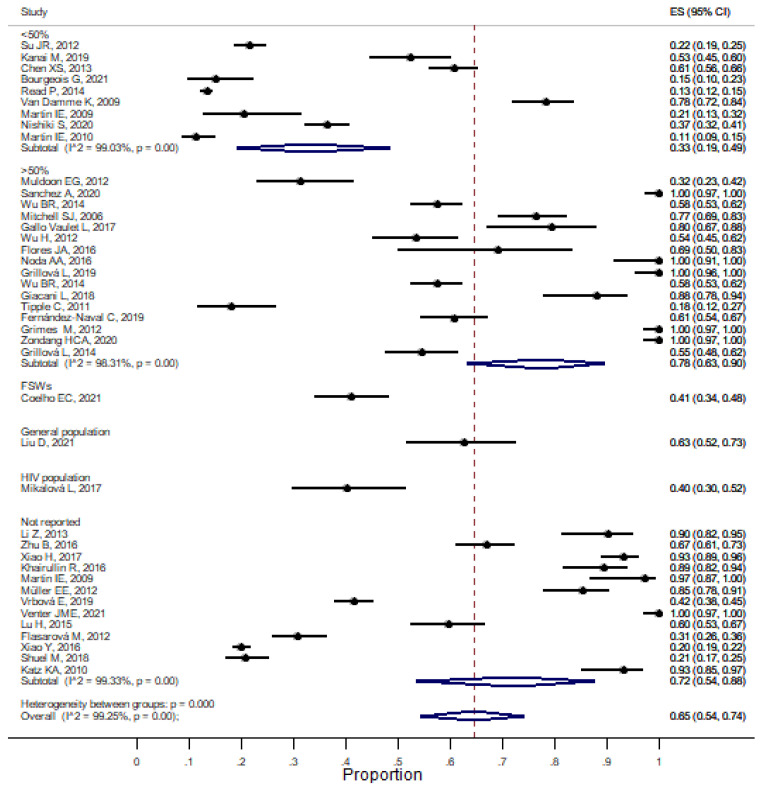
Forest plot of meta-analysis of the molecular detection of syphilis by population type.

**Figure 4 pathogens-11-01546-f004:**
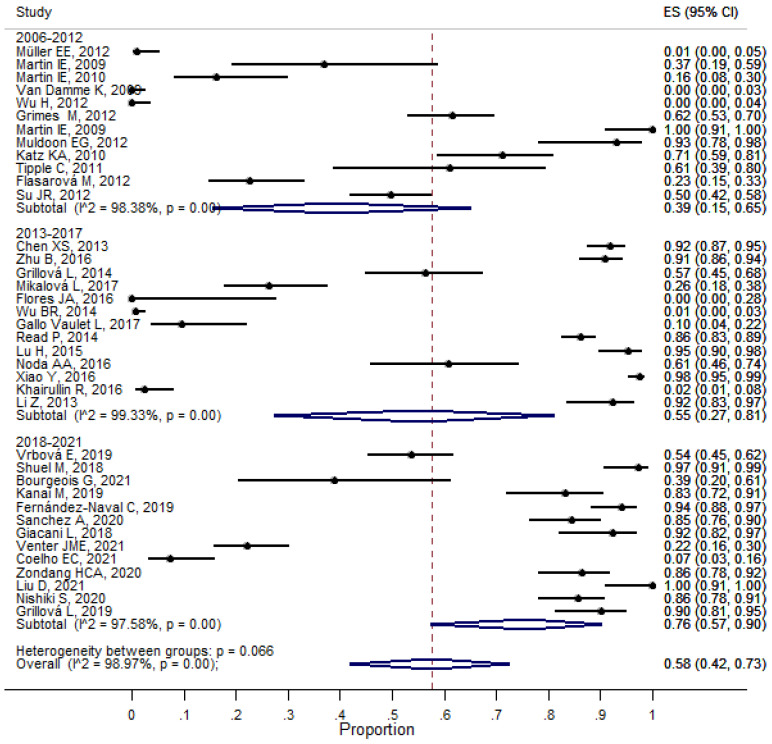
Meta-analysis of proportion mutation A2058G by year period.

**Figure 5 pathogens-11-01546-f005:**
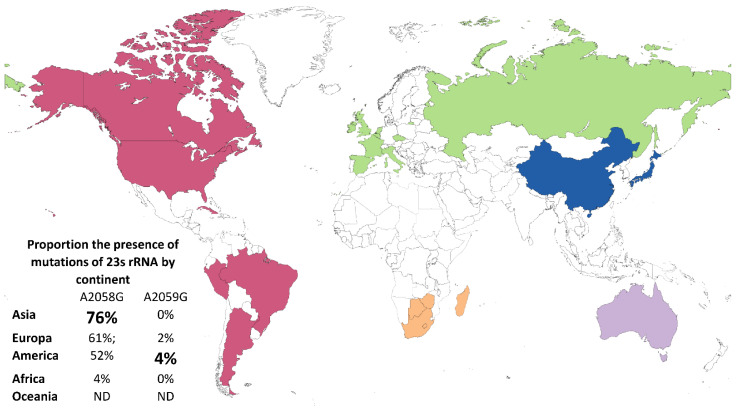
Estimates of the proportion (%) of the mutations of 23s rRNA *T. pallidum* by continent. In solid colors are the observed countries (stratified by continent) in which mutation detection studies were carried out.

**Figure 6 pathogens-11-01546-f006:**
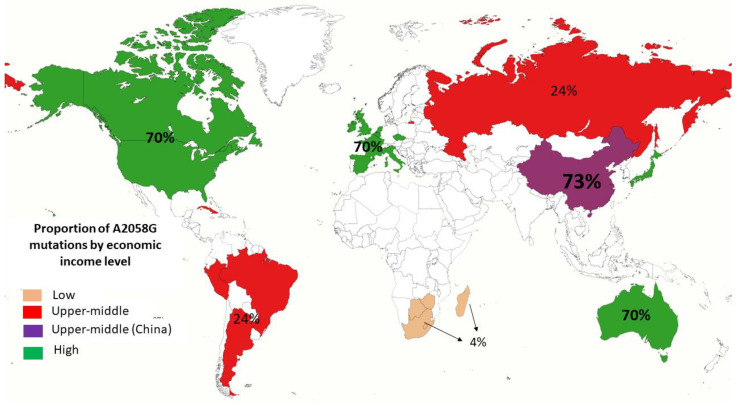
Proportion of A2058G mutation by economic income level. Classified by low, low-middle, upper-middle and high economic income level of the country according to the World Bank.

**Table 1 pathogens-11-01546-t001:** Descriptive characteristics of eligible studies [12,24,25,26,27,28,29,30,31,32,33,34,35,36,37,38,39,40,41,42,43,44,45,46,47,48,49,50,51,52,53,54,55,56,57,58,59,60].

First Author, Publication Year	Continent	Country	Recruiment Year	Sample Size (*n*)	Sex	Age Range or Median/Mean (Years)	MSM Population (*n*)	HIV Population (*n*)	Syphilis Molecular Detection (%)	A2058G Mutation (%)	A2059G Mutation (%)	Previous tx	Strain	Recruiting Place	Income Economies
Bourgeois G, 2021 [26]	Europe	France	2018	119	Both	NR	6	NR	13.95	39	NE	NO	NE	STI clinic	High
Mitchell SJ, 2006 [27]	America	USA	2000–2004	154	Men	30.5–44	138	37	76.6	NE	NE	Yes	NE	STI clinic	High
Read P, 2014 [28]	Oceania	Australia	2004–2011	3037	Men	21–71	203	118	13.5	86	NE	Yes	Yes	STI clinic	High
Fernández-Naval C, 2019 [29]	Europe	Spain	2015	213	Both	NR	202	NR	61	99.1	0.9	NR	14d	STI clinic	High
Giacani L, 2018 [30]	Europe	Italy	2016–2017	60	W (*n* = 1); M (*n* = 59)	18–70	52	21	88.3	98	2	NR	14d	STI clinic	High
Nishiki S, 2020 [31]	Asia	Japan	2017–2018	482	Both	<15–>60	34	NR	36.5	100	0	Yes	14d	STI clinic	High
Vann Damme K, 2009 [32]	Africa	Madagascar	2000–2007	186	Both	18–53	4	NR	78.5	0	NE	Yes	NE	STI clinic	Low
Wu H, 2012 [33]	Asia	China	2009–2011	136	W (*n* = 4); M (*n* = 132)	19.6–57.9	109	93	55.3	0	0	NO	14f	HIV care clinic	Upper- middle
Noda AA, 2016 [12]	America	Cuba	2012–2015	41	M (*n* = 41)	16–52	32	28	100	61	0	Yes	14f	General care clinic	Upper- middle
Lu H, 2015 [34]	Asia	China	2007–2009	182	W (*n* = 34); M (*n* = 75)	39.9 Mean	NR	NR	59.9	95.4	NE	Yes	NE	STI clinic	Upper- middle
Chen XS, 2013 [35]	Asia	China	2008–2011	391	W (*n* = 50); M (*n* = 161)	29–44	2	1	60.9	91.9	NE	Yes	NE	STI clinic	Upper- middle
Tipple C, 2011 [36]	Europe	UK	2006–2008	99	Men	24–54	17	11	19	91	8.3	NR	14d	STI clinic	High
Zhu B, 2016 [37]	Asia	China	2012–2014	265	W (37); M (141); n/d 87)	17–85	NR	5	100	91	NE	Yes	14d	STI clinic	Upper- middle
Martin IE, 2009 [38]	America	Canada	2007–2008	39	W (*n* = 6); M (33)	21–57	NR	1	97.4	100	NE	NR	14f	STI clinic	High
Müller EE, 2012 [39]	Africa	South Africa and Lesotho	2005–2010	117	M (*n* = 117)	NR	NR	NR	85	1	NE	Yes	14d	STI clinic	Low
Vrbová E, 2019 [40]	Europe	Czech Republic	2004–2017	675	W (*n* = 25); M (*n* = 244); n/d (*n* = 406)	0–71	NR	NR	41.6	53.7	4.4	NR	NE	STI clinic	High
Liu D, 2021 [25]	Asia	China	2016–2017	78	W (*n* = 8); M (*n* = 41); n/d (*n* = 29)	27–62	NR	NR	62.8	100	0	NR	16d	NR	Upper- middle
Martin IE, 2009 [38]	America	Canada	2007–2008	68	W (*n* = 21); M (*n* = 47)	0–97.3	3	NR	50.6	36.8	NE	NR	NE	STI clinic	High
Zondag HCA, 2020 [41]	Europe	Nederland	2016 y 2017	135	W (*n* = 1); M (*n* = 134)	33–50	74	29	100	86	0	Yes	14d	Health Public Lab	High
Katz KA, 2010 [42]	America	USA	2004–2007	74	M (*n* = 69); n/d (*n* = 5)	23–64	NR	30	93	71.2	NE	Yes	14d	STI clinic	High
Xiao Y, 2016 [43]	Asia	China	2013–2015	2253	W (*n* = 1243); M (*n* = 1010)	16.1–82.5	NR	NR	20.2	97.5	0	NR	14d	STI clinic	Upper- middle
Kanai M, 2019 [44]	Asia	Japan	2017	156	W (*n* = 12); M (*n* = 61); n/d (*n* = 83)	NR	25	NR	52.6	83.3	0	NR	14d	STI clinic	High
Shuel M, 2018 [45]	America	Canada	2012–2016	354	W (*n* = 14); M (*n* = 60); n/d (*n* = 280)	1 month–70 years	NR	NR	20.9	97.3	2.7	NR	14d	STI clinic	High
Mikalová L, 2017 [24]	Europe	Belgium	2014–2015	72	Both	31–50	NR	72	40.3	65.5	0	NR	14j	STI clinic	High
Grillová L, 2014 [46]	Europe	Czech Republic	2011–2013	188	W (*n* = 22); M (*n* = 166)	35.4	96	37	54.8	56	10.1	NR	NE	STI clinic	High
Gallo Vaulet L, 2017 [47]	America	Argentina	2006 y 2013	54	W (*n* = 6); M (*n* = 48)	17–66	27	12	76.9	9	4.7	NR	14d	General care clinic	Upper- middle
Martin IE, 2010 [48]	America	Canada	2007–2009	374	W (*n* = 146); M (*n* = 177); n/d (*n* = 51)	0.1–97	7	NR	11.5	16.3	NE	NR	14d	General care clinic	High
Wu BR, 2014 [49]	Asia	China	2009–2013	375	W (*n* = 1); M (*n* = 371); n/d (*n* = 3)	31.8	349	306	57.6	7	0	NR	14f	HIV care clinic	Upper- middle
Grillová L, 2019 [50]	America	Cuba	2012–2017	83	Both	25–44	69	57	100	90.3	NE	Yes	NE	STI clinic	Upper- middle
Xiao H, 2017 [51]	Asia	China	2014–2015	183	Both	NR	NR	NR	93.4	NE	NE	NR	NE	STI clinic	Upper- middle
Su JR, 2012 [52]	America	USA	2007–2009	651	W (*n* = 21); M (*n* = 118); n/d (*n* = 512)	13–68	82	28	21.6	53.2	NE	NR	14d	STI clinic	High
Flasarová M, 2012 [53]	Europe	Czech Republic	2004–2010	294	W (*n* = 35); M (*n* = 156); n/d (*n* = 103)	30.4	NR	NR	30.9	60.3	39.3	NR	14d	STI clinic	High
Sanchez A, 2020 [54]	Europe	France	2010–2017	146	W (*n* = 3); M (*n* = 143)	21–71	115	39	100	85	0	Yes	NE	STI clinic	High
Wu BR, 2014 [49]	Asia	China	2009–2013	375	W (*n* = 1); M (*n* = 371); n/d (*n* = 3)	NR	349	306	57.6	7	0	NR	NE	NR	Upper- middle
Khairullin R, 2016 [55]	Asia	Russia	2013–2014	95	W (*n* = 54); M (*n* = 41)	15–80	NR	0	89.5	2.4	NE	Yes	14d	STI clinic	Upper- middle
Muldoon EG, 2012 [56]	Europe	Ireland	2009–2010	92	W (*n* = 4); M (*n* = 88)	19–64	72	NR	31.5	93.1	NE	NR	NE	STI clinic	High
Coelho EC, 2021 [23]	America	Brazil	2015–2019	180	W (180)	23.5	0	NR	41.1	7.4	8.8	Yes	NE	STI clinic	Upper- middle
Venter JME, 2021 [57]	Africa	Botswana, Zimbabwe, and South Africa	2008–2018	135	Both	24–34	NR	46	100	22	0	NR	14d	STI clinic	Low
Flores JA, 2016 [58]	America	Peru	2013–2014	26	Both	NR	26	12	69.2	0	0	NR	14d	STI clinic	Upper- middle
Li Z, 2013 [59]	Asia	China	2010–2012	73	Both	NR	NR	0	90.4	92.4	7.6	NR	NE	STI clinic	Upper- middle
Grimes M, 2012 [60]	America	USA	2001–2010	129	W (*n* = 2); M (*n* = 127)	NR	125	112	100	62	1	Yes	14d	General care clinic	High

General characteristics of eligible studies, NR: Not reported; NE: Not evaluated; W: women, M: men, *n*: sample size; %: proportion; tx: treatment; STI: Sexual Transmission Infection; HIV: human immunodeficiency virus.

**Table 2 pathogens-11-01546-t002:** Assessment of the quality of eligible studies [12,23,24,25,26,27,28,29,30,31,32,33,34,35,36,37,38,39,40,41,42,43,44,45,46,47,48,49,50,51,52,53,54,55,56,57,58,59,60].

Study	Study Design	Selection (5)	Comparability (2)	Outcome (3)	Global Punctuation (10)	Interpretation
Bourgeois G, 2021 [26]	Cross-sectional	4	0	3	7	High
Mitchell SJ, 2006 [27]	Cross-sectional	3	0	3	6	Moderate
Read P, 2014 [28]	Cross-sectional	3	0	3	6	Moderate
Fernández-Naval C, 2019 [29]	Cross-sectional	4	0	3	7	High
Giacani L, 2018 [30]	Cross-sectional	4	0	2	6	Moderate
Nishiki S, 2020 [31]	Cross-sectional	4	2	2	8	High
Damme K, 2009 [32]	Cross-sectional	5	0	3	8	High
Wu H, 2012 [33]	Cross-sectional	4	0	3	7	High
Noda AA, 2016 [12]	Cross-sectional	4	0	2	6	Moderate
Lu H, 2015 [34]	Cross-sectional	2	2	3	7	High
Chen XS, 2013 [35]	Cross-sectional	4	2	3	9	High
Tipple C, 2011 [36]	Cross-sectional	4	0	3	7	High
Zhu B, 2016 [37]	Cross-sectional	2	2	3	7	High
Martin IE, 2009 [38]	Cross-sectional	4	0	2	6	Moderate
Müller EE, 2012 [39]	Cross-sectional	5	0	3	8	High
Vrbová E, 2019 [40]	Cross-sectional	4	2	3	9	High
Liu D, 2021 [25]	Cross-sectional	4	0	2	6	Moderate
Martin IE, 2009 [38]	Cross-sectional	2	0	3	5	Moderate
Zondag HCA, 2020 [41]	Cross-sectional	2	0	3	5	Moderate
Katz KA, 2010 [42]	Cross-sectional	4	2	3	9	High
Xiao Y, 2016 [43]	Cross-sectional	2	0	3	5	Moderate
Kanai M, 2019 [44]	Cross-sectional	4	2	3	9	High
Shuel M, 2018 [45]	Cross-sectional	3	2	2	7	High
Mikalová L, 2017 [24]	Cross-sectional	3	2	2	7	High
Grillová L, 2014 [46]	Cross-sectional	3	2	3	8	High
Gallo Vaulet L, 2017 [47]	Cross-sectional	4	2	3	9	High
Martin IE, 2010 [48]	Cross-sectional	3	2	2	7	High
Wu BR, 2014 [49]	Cross-sectional	3	2	3	8	High
Grillová L, 2019 [50]	Cross-sectional	4	2	3	9	High
Xiao H, 2017 [51]	Cross-sectional	4	0	2	6	Moderate
Su JR, 2012 [52]	Cross-sectional	3	2	3	8	High
Flasarová M, 2012 [53]	Cross-sectional	3	2	3	8	High
Sanchez A, 2020 [54]	Cross-sectional	4	0	2	6	Moderate
Wu BR, 2014 [49]	Cross-sectional	3	0	2	5	Moderate
Khairullin R, 2016 [55]	Cross-sectional	4	0	2	6	Moderate
Muldoon EG, 2012 [56]	Cross-sectional	3	0	2	5	Moderate
Coelho EC, 2021 [23]	Cross-sectional	4	2	3	9	High
Venter JME, 2021 [57]	Cross-sectional	4	0	3	7	High
Flores JA, 2016 [58]	Cross-sectional	3	0	2	5	Moderate
Li Z, 2013 [59]	Cross-sectional	4	0	2	6	Moderate
Grimes M, 2012 [60]	Cross-sectional	4	0	2	6	Moderate

Quality assessment of eligible studies. A study had “high quality” with an overall score of 7 stars, “moderate quality” when it reached a range of 4 to 6 stars, and “low quality” when it achieved an overall score of 3 stars.

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
