# Peer review of "Syphilis as Re-Emerging Disease, Antibiotic Resistance, and Vulnerable Population: Global Systematic Review and Meta-Analysis"

_pathogens, 2022, doi:10.3390/pathogens11121546_

Round 1
Reviewer 1 Report
The search criteria for the review related to syphilis/T pallidum or antibiotics and genotyping or resistance mutations. Evaluation was ‘independent’ but it is not stated how many evaluated each paper even though it says discrepancies resolved by discussion. The same applies to data extraction and quality assessment. The research questions were quite general and some not involving genotyping so may not be covered by the searches e.g. antibiotic resistance, re-emerging disease. A relatively small number of articles were selected but the criteria used required molecular detection and antibiotic resistance confirmed with PCR. Quality assessment was moderate to high for eligible studies.
3.3 The proportions diagnosed with syphilis by molecular detection are a selected population as mostly from STI clinic. There was a variety of income groups and countries as well as documented behaviours. This should be made clear including in the abstract where the meaning may be misinterpreted.
Discussion – were any efforts made to access original data from studies to determine prevalence in MSM subpopulations? It is assumed that overall prevalence in studies with higher proportions of MSM indicated that MSM was a risk factor. Abuse of antibiotics and emergence of resistance are important globally. The authors should discuss the mechanisms or resistance as this is a finding with macrolides only in their report.
The English needs attention e.g. p1line 29 p2 line 53, 68, 72, p9 line 372.
Author Response
México, 07 Diciembre 2022
Dear Reviewer,
We reply to the comments suggested in our work, hoping that the modifications made are satisfactory for publication.
Response to Reviewer 1 Comments
Point 1: a)The search criteria for the review related to syphilis/T pallidum or antibiotics and genotyping or resistance mutations. b)Evaluation was ‘independent’ but it is not stated how many evaluated each paper even though it says discrepancies resolved by discussion. The same applies to data extraction and quality assessment. c)The research questions were quite general and some not involving genotyping so may not be covered by the searches e.g. antibiotic resistance, re-emerging disease. d)A relatively small number of articles were selected but the criteria used required molecular detection and antibiotic resistance confirmed with PCR. Quality assessment was moderate to high for eligible studies.
Response 1: a) The search logarithm included works that were related to sífillis/T. pallidum, but also detected antibiotic resistance. There was a finger error in the logarithm which has been corrected, as follows:
“Preliminary studies were selected using the following search algorithm: (“syphilis”, OR “T. pallidum” OR “Treponema pallidum” OR “prevalence of syphilis” AND “MSM” OR “Men sex men”AND “heterosexual” AND “pregnant” AND “PLWH” OR “people living with HIV” AND “penicillin” OR “penicillin g benzathine” AND “doxycycline” OR “azithromycin” OR “ampicillin” AND “macrolides” AND “tetracycline” AND “genotyping” AND molecular characterization” AND “PCR” AND “mutations of resistance to antibiotics” OR “A2058G” AND “A2059G”. LINE (92-98).
- b) The activities carried out by the authors are identified by the initials of the authors in the lines: 100-103, 117-119 and 131-133.
- c) Thank you for your comment. The main search criteria of this study was the identification of pallidum done by PCR, since it is a first step for the detection of point mutations by molecular techniques. Research questions were modified to avoid confusión: “1) There is an increase of syphilis worldwide?, 2) Are the MSM the core group population in syphilis transmission?, 3) Is there published evidence about point mutations that confer antibiotic resistance in T. pallidum as a causal agent of syphilis?” (Line 104-106).
- d) Thank you for your comment. Although there are several studies that address the detection by PCR of pallidum and the point mutations of resistance to antibiotics, only these 41 studies that we included in our study met the criterion that both variables will be identified in the study; in addition to meeting the eligibility criteria set out in the methodology section: “1) literature published in English and Spanish language that was made only in humans and on any date, 2) original observational studies (prospective and retrospective), 3) molecular detection of T. pallidum DNA and antibiotic resistance confirmed with PCR assay in the same article, 4) excluded gray literature, and 5) excluded abstract and full-text not available (line 110-114)”.
Point 2: 3.3 The proportions diagnosed with syphilis by molecular detection are a selected population as mostly from STI clinic. There was a variety of income groups and countries as well as documented behaviours. This should be made clear including in the abstract where the meaning may be misinterpreted.
Response 2: Thanks for the suggestion. Because of the restriction in the number of words in the abstract it was not possible for us to include this point, however, the following text was added in the conclusion: “Although this work does not focus on any characteristic about the place of recruitment of the population, most of the eligible studies were conducted in STI clinics. More information is needed on the dynamics of this disease in the general population.” (line 399-401)
Point 3: Discussion – were any efforts made to access original data from studies to determine prevalence in MSM subpopulations? It is assumed that overall prevalence in studies with higher proportions of MSM indicated that MSM was a risk factor. Abuse of antibiotics and emergence of resistance are important globally. The authors should discuss the mechanisms or resistance as this is a finding with macrolides only in their report.
Response 3: In studies that did not give a specific characteristic of the type of population, the authors were contacted and asked for information on the percentage and/or the number of MSM population participating in their study. We found important information on the high prevalence of syphilis in the MSM population that should be taken into account as a risk factor.
We agree that antibiotic abuse and emergence of resistance is a global health problem. In syphilis infections, many authors have published data about point mutations in 23S rRNA gene (A2058G and A2059G) that confers resistance to macrolides (all the papers included in our work); especially azithromycin. However, part of the limitations of this work is that the consumption of macrolides meta-analysis could not be addressed, since the studies did not have the information, so the analyses were performed with the prevalence of mutations (line 372-374).
The English needs attention e.g. p1line 29 p2 line 53, 68, 72, p9 line 372.
Proof of revision of the English language is attached.
Best regards,
Citlalli Orbe
Reviewer 2 Report
This is may become an interesting review that looks mainly at the antibiotic resistance associated with syphilis, although macrolides and tetracyclines are seldom used in the therapy of syphilis.
1. The re-emergence of syphilis should not be stated in the title because the authors can only report the increase in the incidence of the disease, but they do not have the authority to declare the re-emergence, nor an epidemic (line 294, 370).
2. The difference between the serological diagnostic tests for syphilis is not the cost, but above all the specificity and the time interval of the positivity. That is why there is a logical sequence in the use of non-treponemic and, respectively, treponemic tests. Please clarify accordingly (lines 55-62).
3. Line 61. PCR may be used only in early or congenital syphilis.
4. Line 63. Inexact. Clarify.
5. Line 66. Please, provide examples of such countries. Are studies on the population of these countries included in the review?
6. Line 71. Provide more data on these mutations.
7. Table 1. Include: stage of syphilis and specimen used for molecular detection.
8. Line 211. VIH?
9. Lines 216-217. Provide percentage and proportion (in brackets).
10. Line 232. What is the number of participants in these studies? Is the incidence of syphilis not higher in studies on larger groups of patients, regardless of the participation of MSM in the group?
11. Figures 4 and 5. Please, clarify the legend and colors.
12. Lines 334-337. I don’t agree. Furthermore, reference 66 is old. See reference 12.
13. Line 365. Molecular methods do not represent the gold standard of syphilis diagnosis. Its sensitivity depends on the concentration of bacteria in the clinical sample. The text should be modified. Of course, only molecular methods can detect resistance genes.
14. Data on the frequency of use of macrolides/tetracyclines in syphilis therapy should be added
15. Bacteria names should be written in italics.
16. The format of the bibliography should be revised, as well as the mode of citation in the text. There are few recent references.
17. The punctuation marks must be checked. The manuscript is drafted carelessly.
18. The manuscript is written in poor English; therefore, it is difficult to follow. It contains many grammatical and syntax errors of the English language and must be checked from this point of view and corrected by an authorized person.
19. There are variations in the writing font for the abstract.
Author Response
México, 07 Diciembre 2022
Dear Reviewer,
We reply to the comments suggested in our work, hoping that the modifications made are satisfactory for publication.
Response to Reviewer 2 Comments
Point 1: This is may become an interesting review that looks mainly at the antibiotic resistance associated with syphilis, although macrolides and tetracyclines are seldom used in the therapy of syphilis.
Response 1: Thank you for this comment. There is information about the use of other antibiotic treatments for syphilis. Mostly in penicillin allergies the recommendation is doxycycline, and there is evidence about efficacy of azithromycin in syphilis infections (https://scielo.isciii.es/scielo.php?script=sci_arttext&pid=S0212-71992002000200010, https://www.ncbi.nlm.nih.gov/pmc/articles/PMC4545322/, https://www.ncbi.nlm.nih.gov/pmc/articles/PMC6254479/). To date, azithromycin is recommended in 16 countries (Review of national treatment guidelines for sexually transmitted infections in the Western Pacific Region; Report No.: WPR/2018/DCD/004. Available from: https://apps.who.int/iris/handle/10665/279732); however, we are aware of the current CDC guidelines recommending the use of penicillin for syphilis infections (Workowski KA, Bachmann LH, Chan PA, Johnston CM, Muzny CA, Park I, et al. Sexually Transmitted Infections Treatment Guidelines, 2021. MMWR Recomm Rep. 2021 Jul 23;70(4):1–187).
Point 2:The re-emergence of syphilis should not be stated in the title because the authors can only report the increase in the incidence of the disease, but they do not have the authority to declare the re-emergence, nor an epidemic (line 294, 370).
Response 2: Thank you very much for your comment. In our study, we did not address the incidence of syphilis cases since we are not identifying new cases, we evaluated molecular prevalence of syphilis. According to the evidence found in this meta-analysis, we observed an increase in people diagnosed with syphilis and suggested it “could become a re-emergent disease". The suggestion is placed since as well mentioned we have no authority to declare a state of re-emergence, so the following modification is made in the text to avoid future confusion: “Therefore, we observed and increased of cases of syphilis, and we suggested that this disease could become a re-emerging disease”.(Line 331-333). And the following sentence was changed: “there are a set of characteristics that could be causing the increase of…” (line 393). We agreed that we do not have the authority to declare a re-emergence event. However, in accordance with other authors we present data and evidence about the rising of syphilis cases. Syphilis is an infection that was believed to be eliminated and has currently shown an increase in cases in different populations (such as pregnant women, newborns, and MSM), and clinical presentations. Taking into account that re-emergence are defined by the reappearance and increase in the number of infections of an already known pathology that, due to the few cases registered, was no longer considered a public health problem; like other authors and publications, we respectfully allow ourselves to use the term re-emergence to describe our finding.
(https://pubmed.ncbi.nlm.nih.gov/22963062/; https://pubmed.ncbi.nlm.nih.gov/32762856/; https://pubmed.ncbi.nlm.nih.gov/29022569/; https://www.ncbi.nlm.nih.gov/pmc/articles/PMC6089383/; https://www.ncbi.nlm.nih.gov/pmc/articles/PMC5354565/)
Point 3: The difference between the serological diagnostic tests for syphilis is not the cost, but above all the specificity and the time interval of the positivity. That is why there is a logical sequence in the use of non-treponemic and, respectively, treponemic tests. Please clarify accordingly (lines 55-62).
Response 3: Thanks for the observation, the paragraph with respect to diagnostic has been modified to clarify according to the suggestion “The diagnosis of syphilis is based on clinical symptoms and confirmatory laboratory tests. There are treponemal tests (i.e., ImmunoSorbent-ELISA Linked Enzyme Assay) and nontreponemal tests (NT, i.e., Venereal Disease Research Laboratory test-VDRL and Rapid Plasma Reagin-RPR). Although The nontreponemal tests had been routinely used because they are simple to operate, inexpensive and rapid although the sensibility is relatively low in the diagnosis of syphilis and depending on the stages (62-78 % in primary syphilis, 97-100% for secondary syphilis, and 82-100% for early latent syphilis). In case of treponemal tests the sensibility and specificity are better to non-treponemal tests (100% in secondary syphilis, 95.2–100% sensitive in early latent syphilis, and 86.8–98.5% sensitive in late latent syphilis). However, detection of antibodies by treponemal test difficult the evaluation of treatment efficacy and there is not possible to distinguish between active stage and previously treated infection (Luo et al 2021). Recently, the CDC approved the use of polymerase chain reaction (PCR) for testing of T. pallidum on suspected primary lesions, as one of recommended tests. PCR´ specificity and sensibility have been major (96.6% and 78.4%, respectively) in cases of primary syphilis respect to clinical diagnosis, and better to serology, because not require the presence of antibodies or large numbers of bacterias for microscopy, and are economical and practical, the cross-reaction reagin antibody is not very specific in detecting syphilis; on the other hand, treponemal tests are expensive, time-consuming and technically difficult. Molecular techniques such as PCR have been suggested to be complementary techniques in the detection of syphilis ” (line 54-71).
Point 4: Line 61. PCR may be used only in early or congenital syphilis.
Response 4: Thanks for the comment, however, recent information indicates that PCR may be a tool that aids in the diagnosis of syphilis; according to the Sexually Transmitted Infections Treatment Guidelines, 2021 of CDC, it indicates that when serological tests and clinical tests do not correct, PCR can be used as confirmatory evidence (Workowski KA et al 2021 MMWR Recomm). In addition, Shukalek CB, et al obtained a specificity of 99.9% and sensitivity of 49.5% for the detection of T. pallidum by PCR, and indicate that serology has low yields and needs multiple tests, and often molecular tests can provide support in the diagnosis of primary syphilis and can be used as a confirmatory test (Shukalek CB, et al 2021 Front Cell Infect Microbiol).
Point 5: Line 63. Inexact. Clarify.
Response 5: The text was changed this way “According to WHO, use of macrolides and tetracycline as an alternative treatment for syphilis has increased ” (line 72-73).
Point 6: Line 66. Please, provide examples of such countries. Are studies on the population of these countries included in the review?
Response 6: The text was modified in the lines 74-75. Included only 1 study that was conducted in Australia, so we do not believe that the results obtained are impacted by the situation of stock-outs of BPG.
Point 7: Line 71. Provide more data on these mutations.
Response 7: The text was changed this way “epidemiological evidence with respect to syphilis reemergence and the increase in AMR from A2058 G and A2059G point mutations worldwide” (line 81-83)
Point 8: Table 1. Include: stage of syphilis and specimen used for molecular detection.
Response 8: In order to not alter the contents of Table 1, a new table was added which is called "Table S2. Continuity of descriptive characteristics of eligible studies". In this we detail the type of sample and the stage that were reported in each study included in this work. And the following text was added “Most of the studies (37/41, 90%) used lesion samples for the molecular detection of T. pallidum and point mutations, and all studies reported patients with primary syphilis (Table S2). (line 253-255)”
Point 9: Line 211. VIH?
Response 9: Thanks for the observation, in Table 1 was changed VIH for "HIV"
Point 10: Lines 216-217. Provide percentage and proportion (in brackets).
Response 10: Corrected.
Point 11: Line 232. What is the number of participants in these studies? Is the incidence of syphilis not higher in studies on larger groups of patients, regardless of the participation of MSM in the group?
Response 11: Regarding the question What is the number of participants in these studies?: In the section “Materials and Methods/data extraction” the following text was change “The group MSM was categorized as more than 50% (>50%) and less than 50% (<50%) of the total population that participated in each study. (line 153-154)” and in the line 244 and 245 “Subsequently, we stratified by type of population, and we observed that the higher the percentage of the MSM population according to the total of participants in the study” was changed, to avoid future confusion. The number of participants in each study is detailed in Table 1, the fifth column reports the sample size "Sample size (n)" of the included studies. In this work, as mentioned above, we are not addressing new cases of syphilis (incidence), but we are including studies that report "molecular prevalence of syphilis". Regarding the question “Is the incidence of syphilis not higher in studies on larger groups of patients, regardless of the participation of MSM in the group?” , as we can see in Table 1 columns 5 (Sample size (n)), column 8 (MSM population (n)) and column 10 (Syphilis molecular detection (%)), it is observed that the percentage of people diagnosed with syphilis is independent of the sample size and the HSH population. Furthermore, the molecular prevalence of syphilis in MSM patients was independently on board. It should be noted that to avoid a bias in the calculation of prevalence/proportions, the Freeman-Tukey double arcsine transformation approach was used to normalize outcomes before pooling (Nyaga, V.N et al 2004, Arch Public Health). The following text was added “….., using Freeman-Tukey double arcsine transformation” (Line 142-143).
Point 12: Figures 4 and 5. Please, clarify the legend and colors.
Response 12: Tank you for this observation. The quality of figures 4 and 5 was improved.
Point 13: Lines 334-337. I don’t agree. Furthermore, reference 66 is old. See reference 12.
Response 13: Thanks for the suggestion. The text in the lines 354-359 was corrected and a reference actually was added.
Point 14: Line 365. Molecular methods do not represent the gold standard of syphilis diagnosis. Its sensitivity depends on the concentration of bacteria in the clinical sample. The text should be modified. Of course, only molecular methods can detect resistance genes.
Response 14: Thanks for your comment, we agree that the detection of the bacteria depends on the concentration of this in the sample and that the only way to detect mutations is by molecular techniques, so, identifying T. pallidum by PCR, is a tool to assist in the diagnosis of syphilis. In addition to the fact that it is quality control for the detection of mutations and thus corroborate that there is no false positive test, so, the aim of this work is to include studies where the detection of T. pallidum and the mutations A2058 and A2059G was carried out by molecular tests. We do not want to demerit the serological tests, and we do not mention that the molecular tests are a gold standard of syphilis detection, as already mentioned in point 4, the PCR can be used to assist in the diagnosis of syphilis. Line 368-373 was modified as follows, to avoid future confusion: “The strength of our work was the inclusion of eligible studies that performed only the direct detection of T. pallidum and antibiotic resistance mutations by molecular methods, Shukalek CB, et al indicated that the PCR has a specificity of 99.9% and sensitivity of 49.5% for the detection of T. pallidum (70), which allows an objective evaluation since most studies confirmed the diagnosis of syphilis with molecular tests ”
Point 15: Data on the frequency of use of macrolides/tetracyclines in syphilis therapy should be added
Response 15: No data on treatment frequencies with macrolides/tetracyclines, but according of CDC´s support use of alternatives to penicillin in treating primary and secondary syphilis are limited. However, multiple therapies might be effective for nonpregnant persons with penicillin allergy who have primary or secondary syphilis. Doxycycline (100 mg orally 2 times/day for 14 days) and tetracycline (500 mg orally 4 times/day for 14 days) have been used for years and can be effective. Ceftriaxone (1 g daily either IM or IV for 10 days) is effective for treating primary and secondary syphilis; however, the optimal dose and duration of ceftriaxone therapy have not been defined. Azithromycin as a single 2-g oral dose has been effective for treating primary and secondary syphilis among certain populations (Workowski KA 2021). This its relationated with the poin 13.
Point 16: Bacteria names should be written in italics.
Response 16: Corrected
Point 17: The format of the bibliography should be revised, as well as the mode of citation in the text. There are few recent references.
Response 17: Thanks for the comment. Added some references from 2021-2022, however 41 of the cited references cannot be changed, as they are part of the systematic review and meta-analysis; so we consider incorporating the most important and relevant references.
Point 18: The punctuation marks must be checked. The manuscript is drafted carelessly.
Response 18: Proof of revision of the English language is attached.
Point 19: The manuscript is written in poor English; therefore, it is difficult to follow. It contains many grammatical and syntax errors of the English language and must be checked from this point of view and corrected by an authorized person.
Response 19: Proof of revision of the English language is attached.
Point 20: There are variations in the writing font for the abstract.
Response 20: The structure and the letter number and writing source of the article was carried out according to the guidelines and template proposed by the journal.
Best regards,
Citlalli Orbe
Reviewer 3 Report
I am grateful for the opportunity to review the manuscript "Syphilis as re-emerging disease, antibiotic resistance, and vulnerable population: global systematic review and meta-analysis". As well highlighted by the authors, it is very important to establish strategies that help in the diagnosis of syphilis, in the treatment and in the prevention of the increase in antibiotic resistance. Thus, the proposal presented by the authors is interesting. The manuscript was very well designed, executed and written. The findings are useful and should be publicized.
The only request I make is to improve the resolution and size of figure 2 (I suggest placing figures 2A and 2B on top of each other, not as they currently are - side by side).
Author Response
México, 07 Diciembre 2022
Dear Reviewer,
We reply to the comments suggested in our work, hoping that the modifications made are satisfactory for publication.
Response to Reviewer 3 Comments
Point 1: I am grateful for the opportunity to review the manuscript "Syphilis as re-emerging disease, antibiotic resistance, and vulnerable population: global systematic review and meta-analysis". As well highlighted by the authors, it is very important to establish strategies that help in the diagnosis of syphilis, in the treatment and in the prevention of the increase in antibiotic resistance. Thus, the proposal presented by the authors is interesting. The manuscript was very well designed, executed and written. The findings are useful and should be publicized.
Response 1: Thank you very much for your comment and support.
Point 2: The only request I make is to improve the resolution and size of figure 2 (I suggest placing figures 2A and 2B on top of each other, not as they currently are - side by side).
Response 2: the quality of figures 2 and 3 is improved.
Best regards,
Citlalli Orbe
Reviewer 4 Report
Journal: Pathogens - MDPI
Article: Syphilis as Re-emerging Disease, Antibiotic Resistance, and Vulnerable Population: Global Systematic Review and Meta-analysis.
Authors: Yaneth Citlalli Orbe-Orihuela, Miguel Ángel Sánchez-Alemán, Adriana Hernández-Pliego, Claudia Victoria, Medina-García1 and Dayana Nicté Vergara-Ortega.
The article is a review on antibiotic resistance of re-emerging syphilis, in the perspective of vulnerable population. Re-emerging diseases and antibiotic resistance are both main interest topics for the public health, nowadays.
The introduction is enough explaining the context of the subject.
The methodology is accurate described, according to the guideline of systematic reviews, and meta-analyses.
According to the title, the results and conclusions should point the differences between the antibiotic resistance in different vulnerable groups, comparative with general population.
I recommend adding in the discussion section a comment about the role of antibiotic treatment to asymptomatic contacts of early syphilis and the development of antibiotic resistance, considering the new data, published this year:
Denman J, Hodson J, Manavi K. Infection risk in sexual contacts of syphilis: A systematic review and meta-analysis. J Infect. 2022 Jun;84(6):760-769. doi: 10.1016/j.jinf.2022.04.024. Epub 2022 Apr 18. PMID: 35447230.
Line 11- missing dot and space
Line 44 – delete the wrong dot
Author Response
México, 07 Diciembre 2022
Dear Reviewer,
We reply to the comments suggested in our work, hoping that the modifications made are satisfactory for publication.
Response to Reviewer 4 Comments
Point 1: The article is a review on antibiotic resistance of re-emerging syphilis, in the perspective of vulnerable population. Re-emerging diseases and antibiotic resistance are both main interest topics for the public health, nowadays.
Response 1: Thank you for your comment and support.
Point 2: The introduction is enough explaining the context of the subject.
Response 2: Thank you for your comment.
Point 3: The methodology is accurate described, according to the guideline of systematic reviews, and meta-analyses.
Response 3: Thank you very much for your comment and support.
Point 4: According to the title, the results and conclusions should point the differences between the antibiotic resistance in different vulnerable groups, comparative with general population.
Response 4: Thanks for the comment, the objective of this study was to obtain a picture of the dynamism of syphilis in the general population, so, the selection of studies was not restricted to a characteristic of the population, However, most of the eligible studies had a high percentage of HSH population, which tells us that there is a high frequency of T. pallidum infection in MSM population.
Point 5: I recommend adding in the discussion section a comment about the role of antibiotic treatment to asymptomatic contacts of early syphilis and the development of antibiotic resistance, considering the new data, published this year:
Response 5: Thank you very much for the comment, in the line 392-396 we discuss the possible role of postexposure Prophylaxis and Pre-Exposure Prophylaxis for STI Prevention and development of antibiotic resistance. About the article you suggested (Denman J, Hodson J, Manavi K. Infection risk in sexual contacts of syphilis: A systematic review and meta-analysis. J Infect. 2022 Jun;84(6):760-769. doi: 10.1016/j.jinf.2022.04.024. Epub 2022 Apr 18. PMID: 35447230.), unfortunately we could not access it.
Point 6: Line 11- missing dot and space
Response 6: Corrected.
Point 7: Line 44 – delete the wrong dot
Response 7: Corrected.
Best regards,
Citlalli Orbe
Round 2
Reviewer 1 Report
Thank you for revising the paper.
Reviewer 2 Report
Manuscript is improved.